# Sexual and gender identities and alcohol use during the COVID-19 pandemic

**Susan D. Stewart** [ID][1]*, **Wendy D. Manning**[2], **Kristen E. Gustafson** [ID][2], **Claire Kamp Dush** [ID][3]

**1** Department of Sociology and Criminal Justice, Iowa State University, Ames, Iowa, United States of America, **2** Center for Family and Demographic Research and Department of Sociology, Bowling Green State University, Bowling Green, Ohio, United States of America, **3** Minnesota Population Center and Department of Sociology, University of Minnesota, Minneapolis, Minnesota, United States of America

* stewarts@iastate.edu

**Data Availability Statement:** This data set is publicly available and can be found here: https://www.icpsr.umich.edu/web/DSDR/studies/38417.

**Funding:** The funding information is as follows: Eunice Kennedy Shriver National Institute of Child

## Abstract

This study examined differences in alcohol use by sexual and gender identities during the COVID-19 pandemic, and assessed whether variation between groups was explained by pandemic-related stressors and minority stress. Data from 2,429 partnered adults in the National Couples' Health and Time Use Study (n = 3,593) collected from September 2020 to April 2021 were used to model drinking patterns (frequency, amount, and drinking to cope) by sexual and gender identities, COVID-19 stress and disruption, microaggressions, and supportive climate. Regression models indicated differences in drinking by gender and sexual identities, even controlling for demographic and socioeconomic factors. Gay, lesbian, and bisexual people had higher odds of drinking to cope with the pandemic than did heterosexual people, and cisgender men had higher odds than did cisgender women. Gay and lesbian people drank more regularly than did heterosexual people, as did cisgender men in relation to cisgender women. Exclusively bisexual people drank significantly more drinks than exclusively heterosexual people, and cisgender men drank significantly more drinks than did cisgender women and those who identified as trans/another gender identity. COVID-19 stress and minority stress were associated with greater alcohol consumption, but they did not account for these differentials. Moving forward, researchers will need to continuously assess these associations, as sources of discrimination and stress will persist beyond the pandemic. Although LGBTQ+ people have disproportionate sources of stress, they varied in how they used alcohol to cope. Potential sources of resilience among sexual and gender diverse individuals should be explored.

## Introduction

Pandemic-related stressors have taken a toll on our collective social and emotional health [1, 2]. In April 2020, American life satisfaction fell to a 12-year low, and daily stress (up 14%) and worry (up 20%) significantly increased since 2019 [3]. Anxiety, depression, and suicidality also rose [4]. Although life satisfaction has since increased, and stresses and worries have declined, they have yet to return to pre-pandemic levels [3], and alcohol consumption and alcohol-related stress remain elevated [5].

Health and Human Development (1R01HD094081)
to Dr. Wendy D. Manning and Dr. Claire Kamp
Dush, the Minnesota Population Center, University
of Minnesota (P2CHD041023) to Dr. Claire Kamp
Dush, the Center for Family and Demographic
Research (P2CHD050959) to Dr. Wendy D.
Manning, and the Ohio State University
(P2CHD058484) to Dr. Claire Kamp Dush.

**Competing interests:** The authors have declared
that no competing interests exist.

Psychological distress is associated with more frequent use of alcohol, higher volumes of alcohol consumed, and problem drinking [6, 7]. Alcohol consumption and sales in the United States and worldwide have increased since the pandemic [8, 9] as well as drinking to cope [10]. Anxiety, depression, social isolation, boredom, and COVID-19-specific stress were found to be associated with increased alcohol consumption and problems [11–14]. Alcohol-related health emergencies, liver disease, and car accidents have also risen [15–17].

People who identify as LGBTQ+ had significantly higher levels of COVID-19-related anxiety, stress, depression, and loneliness than did cisgender, heterosexual people [18–20]. Yet, most studies of alcohol use during the pandemic did not include measures of sexual and gender identities. This is an important oversight because prior to the pandemic, LGBTQ+ people exhibited higher levels of alcohol use and misuse than did cisgender heterosexual people [21, 22] as well as higher levels of psychological distress, depression, and anxiety [23, 24]. These results are consistent with minority stress theory [25], which posits that those with stigmatized identities face unique, stigma-related stressors that are associated with greater stress, depressive symptoms, and substance use [26–28].

Prior research has found significant variation in alcohol consumption and use of alcohol to cope according to sexual and gender identities [18, 19, 29]. Although this body of work lays the groundwork and is suggestive of differential impacts of the pandemic by sexual and gender identities, this research is largely based on small and/or non-representative samples, excludes minority stress measures, does not include the full array of sexual and gender identities, has limited measures of alcohol consumption, and does not specifically examine the use of alcohol to cope. Results have therefore been mixed. Moreover, although this research took place during the pandemic, most researchers did not assess the effect of specific pandemic-based stressors on alcohol use.

The current study relies on data that are uniquely positioned to assess health behaviors among LGBTQ+ adults, and addresses many of these challenges. The National Couples' Health and Time Study (NCHAT), conducted between September 2020 and April 2021, contains an oversample of people who identify as LGBTQ+, detailed measures of gender and sexual identities, indicators of LGBTQ+ and COVID-19 stressors, and several measures of alcohol use [30]. Our study has three goals. First, we examine variation in patterns of alcohol consumption (use of alcohol to cope, regularity of drinking, and amount consumed) between LGBTQ+ groups and in relation to people who identify as cisgender heterosexual. Given the greater than average everyday stress faced by LGBTQ+ people, as well as their higher level of stress during the COVID-19 pandemic, we expected them to engage in higher levels of alcohol use and be more likely to rely on alcohol to cope. Second, we assess whether sexual and gender identity disparities may in part be explained by pandemic-related stressors (COVID-19 stress and disruption). Third, we consider whether and to what extent minority stress (microaggressions and a non-supportive climate) accounts for differences in alcohol consumption by sexual and gender identity.

## Background

With the pandemic came increased alcohol consumption on a global scale. However, drinking had been on the rise for decades prior to COVID-19. In 2022, the average number of drinks consumed by Americans in the past week was 4.0, compared to 2.8 in 1996, a 43% increase, and alcohol-related deaths were on the increase [5]. Alcohol overuse is the fourth leading preventable cause of death in the United States [31] and accounts for one-third of all traffic fatalities [32].

The negative effect of alcohol on people's emotional, social, and physical health is well-documented [33, 34]. Over a third (36%) of Americans agreed that drinking has "ever been a

cause of trouble in the family" [5]. Alcohol-related deaths along with suicides and drug over-doses are considered responsible for the notable 2015 reversion in U.S. life expectancy [35, 36]. Life expectancy declined further in 2021 due mostly to COVID-19, but also due in part to continued increases in alcohol-associated deaths [37, 38].

## Sexual and gender identity and alcohol use

The number of people identifying as LGBTQ+ is not inconsequential. According to Gallup, the percentage of the American population that identifies as LGBTQ+ was 7% in 2020 and 20% of adults age 25 and younger identify as LGBTQ+ [39]. Young people are increasingly rejecting the "sex and gender binary" in favor of more fluid sexual and gender identities [40, 41]. The language by which people express their sexual and gender identity is complex and continues to evolve. We use LGBTQ+ or *sexually and gender diverse* to refer to people whose identities are not cisgender and/or heterosexual [42]. We use "gender identity" to refer to a person's self-concept as male and/or female with the understanding that people may identify as a gender different from that they were assigned at birth, or may consider themselves something other than strictly male/man or female/woman (e.g., non-binary, gender nonconforming). We use "sexual identity" to refer to who people are attracted to physically and/or romantically and can include heterosexual, gay, lesbian, bisexual, or something else (e.g., pansexual, asexual). When discussing the research of others, we use the authors' own terminology.

Differences in alcohol consumption between men and women are well known. Compared to women, men consume alcohol more frequently and in greater amounts, and are more likely to have an alcohol use disorder (AUD) and experience more alcohol-related deaths [43–45]. Recently there has been a convergence of women's and men's alcohol use patterns [46, 47], driven by increased alcohol use among women [48]. Women experienced a greater increase in alcohol consumption than did men since the onset of the pandemic [49, 50]. Compared to cisgender people, gender diverse people report greater alcohol consumption, binge drinking, and alcohol-related harms [51, 52]. Studies examining differences among gender diverse groups have produced mixed results [18, 53, 54].

Sociocultural and contextual factors underlie gender differences in alcohol consumption Alcohol consumption is intertwined with constructions and demonstrations of masculinity and femininity [55]. Whereas male drinking is associated with stereotypical, generally positive, male traits such as acting as a leader and a willingness to take risks [56, 57], women who drink experience significantly greater societal disapproval than do men, are considered less feminine, and more sexually available than woman who do not [58–61].

People with sexually diverse identities, and those with same-gender sexual experiences, consume more alcohol and have more alcohol problems than heterosexual people [21, 62, 63]. Sexually diverse populations also experienced greater than average increases in alcohol consumption and using alcohol to cope since the pandemic [29, 53]. Some have offered that nonconformity to traditional gender roles, greater tolerance for substance use, and the importance of bars in socialization, could underlie greater alcohol consumption among gender and sexually diverse populations [64, 65].

## LGBTQ+ stress, supportive climate, and alcohol consumption

Minority stress theory suggests that LGBTQ+ people face unique, stigma-related stressors that may translate into worse mental health and greater substance use [25]. Stressors specific to LGBTQ+ populations include internalized homophobia, stigma, hypervigilance, discrimination, and violence [66]. Minority stress is often experienced in the form of microaggressions or "subtle slights, invalidations, and insults that target individuals because of their actual or

perceived membership in a marginalized group or groups" which are associated with poor socioemotional and physical health [67] and have been linked to greater alcohol consumption [68, 69]. Microaggressions in a health care setting also contribute to higher rates of substance use among LGBTQ+ people [70, 71]. However, a supportive climate (e.g., community resources, feelings of safety, belongingness) has been found to mitigate these negative effects [72, 73].

### Pandemic stress, disruption, and alcohol consumption

COVID-19-related-stress is associated with increased alcohol consumption and drinking to cope. This may be amplified for LGBTQ+ people who during the pandemic had worse mental and physical health. COVID-19-related disruptions (e.g., changes in employment), also more prevalent among LGBTQ+ people [74], are associated with worse mental health and greater substance use [20, 75].

### The present study

LGBTQ+ people tend to drink in greater amounts and have worse socioemotional health than do cisgender heterosexual people. They also face a unique set of stressors, discrimination, and harassment tied to their sexual and gender identity. We acknowledge the diversity of this population, and aimed to examine whether and how patterns of alcohol consumption varied by sexual and gender identities. Given the additive stress associated with LGBTQ+ identities, combined with the increased stress of COVID-19, we expected that LGBTQ+ people would have overall higher levels of alcohol consumption and would be more likely to use alcohol to cope with the pandemic than would cisgender heterosexual people. Specifically, we tested whether differentials in alcohol consumption by sexual and gender identities were in part explained by higher levels of COVID-19 stress and disruption, microaggressions, and a less supportive climate. We control for key sociodemographic indicators known to vary by sexual and gender identity, stress, and alcohol use.

We based our analysis on the National Couples' Time and Health Study (NCHAT), a nationally representative sample of American couples conducted at the height of the COVID-19 pandemic. NCHAT contains a robust sample of LGBTQ+ people, multiple measures of alcohol use, comprehensive and inclusive measures of LGBTQ+ identities, and measures of COVID-19-specific stress and disruptions, microaggressions, social climate. Although several studies have focused on drinking among LGBTQ+ people since the pandemic, to our knowledge no other data incorporates these important elements. We address critical gaps in our knowledge about how individuals with sexual and gender diverse identities navigated stressors associated with the pandemic and provide insight into reasons why this group is using alcohol to cope.

### Method

### Sample

NCHAT is a nationally representative sample of 3,642 partnered respondents with oversamples of sexual and gender diverse persons. Age ranged from 20 to 60 years and the sample was married or living with a cohabiting partner. Respondents were part of the Gallup Panel and data were collected on-line in Spanish and English from September 2020 to April 2021. The study was approved by The Ohio State Behavioral and Social Sciences IRB (#2018B0246) and The University of Minnesota (STUDY00013894). Consent was established at the beginning of the survey. Respondents were shown and reviewed the fully informed consent, and the survey

only continued if positive consent was obtained by selecting "I consent." Given the oversampling, person-level weights were used. The analytic sample of 3,593 includes English and Spanish speakers and was limited to those with valid responses on all items. Analyses of regularity of drinking and number of drinks consumed was limited to 2,429 respondents who reported drinking in the last 30 days.

## Variables

Using alcohol to cope was measured with, "What are you doing to cope with the coronavirus pandemic?" Respondents who selected "drinking alcohol" were coded as 1. Alcohol consumption was measured in two ways: "Now, thinking back over the last 30 days, about how regularly did you drink alcoholic beverages such as wine, beer, or liquor? Would you say that it was: (a) once in the last month; (b) 2 to 3 times a month; (c) 1 or 2 days a week; (d) 3 or 4 days a week; (e) 5 or 6 days a week; (f) once a day; (g) more than once a day?" If respondents indicated that they drank in the last 30 days, they were subsequently asked, "Over the last 30 days, about how many drinks would you have on a typical day when you drank?" ranging from 1 to 20+. We tested the validity of our alcohol measures. Drinking to cope was positively correlated with regularity of drinking and number of drinks consumed. Having a family member, friend, or doctor express worry about the respondent's alcohol consumption was associated with increased odds of drinking to cope, more regular drinking, and drinking in greater amounts.

## Sexual and gender identities

The question used to identify sexual identity in the NCHAT was, "What do you consider yourself to be? Select all that apply," with eleven responses: "heterosexual or straight," "gay or lesbian," "bisexual," "same-gender loving," "queer," "pansexual," "omnisexual," "asexual," "don't know," "questioning," and "something else" with an option to specify. We coded respondents into four mutually exclusive categories: (a) exclusively heterosexual, (b) exclusively gay or lesbian, (c) exclusively bisexual, and (d) pansexual, omnisexual, queer, "another identity," and those who chose more than one identity. The question used to identify gender identity was, "Which of the following best describes your gender?" Responses were "man," "woman," "trans man," "trans woman," and "do not identify as any of the above." We coded gender into three mutually exclusive categories: (a) cisgender man, (b) cisgender woman, and (c) transgender or another gender identity.

## COVID-19 stress and disruption

COVID-19 stress was measured by, "How stressed are you about the following?" and summing their responses to four items: (a) getting coronavirus, (b) my spouse or partner getting coronavirus, (c) my parents, siblings, or other family members getting coronavirus, and (d) giving someone the coronavirus. Item responses ranged from 1 (not at all stressed) to 5 (very stressed; alpha = .87). COVID-19 disruption was measured by, "To what extent has your life been affected or disrupted by the coronavirus situation?" with responses ranging from 1 (not at all) to 4 (a great deal).

## Minority stress

For minority stress, respondents were asked, on a scale ranging from 1 (never) to 5 (very often): "In your day-to-day life over the past months how often did any of the following things happen to you?" followed by: (a) you were treated with less respect than other people, (b) you received poorer service than other people at restaurants or stores, (c) people acted as if they

were afraid of you, (d) people acted as if they thought you were dishonest, (e) people acted as if they were better than you, (f) you were called names or insulted. Responses were summed (alpha = .86). Supportive climate was measured by, "Is the city or area where you live a good place or not a good place to live for. . .?" and summing their responses to two items: (a) people who are gay, lesbian, or bisexual, and (b) people who are transgender or nonbinary (agender, gender-neutral, gender-fluid). Responses ranged from 1 (not a good place) to 5 (a good place; alpha = .90).

### Covariates

Sociodemographic variables included: age (20–29, 30–39, 40–49, 50–60), race/ethnicity (non-Hispanic White, non-Hispanic Black, non-Hispanic Asian, Hispanic/Latinx, non-Hispanic multi-race, and another); education (high school or less, some college, college degree) and employment (not employed, employed but not working, employed part-time, employed full-time), relationship status (married, cohabiting), parenthood status (have children versus not), and month of interview.

### Analysis plan

We first provide descriptive information on alcohol usage among people of different sexual and gender identities. We then used logistic regression models to estimate the odds of using alcohol to cope, OLS regression to estimate regularity of alcohol consumption, and negative binomial models to estimate number of drinks consumed (appropriate when variables are skewed). Analyses of regularity of consumption and number of drinks was limited to respondents who reported drinking in the last 30 days. This was 68% of the total sample which is somewhat higher than the national figure of 52% (results not shown; NIAAA 2023), likely the result of our sample being restricted to those 60 and younger. We estimated four sets of models for each outcome in a stepwise fashion, adding each set of covariates sequentially: (a) sexual and gender identities and month of interview, (b) sociodemographic characteristics, (c) micro-aggressions and supportive climate, and (d) COVID-19 stress and disruption. We tested for mediating effects using the Clogg test, which tests for statistically significant differences in regression coefficients between nested models [76]. Month of interview was not associated with drinking to cope, regularity, or amount consumed (results not shown). All analyses were conducted using STATA17.

### Results

[Table 1] provides a description of the sample. Over one-quarter of respondents reported drinking alcohol as a way to cope with the pandemic. Among respondents who reported drinking in the past 30 days, the average frequency was 1 to 2 days a week. Respondents consumed an average of 2.3 drinks per drinking occasion. The sexual identity of respondents was predominately exclusively heterosexual with small percentages identifying as exclusively gay/lesbian, exclusively bisexual, and multi or another sexual identity. About half of respondents identified as a cisgender man, half identified as a cisgender woman, and less than 1% identified as transgender or another sexual identity. These statistics were similar across samples of people who drank in the last 30 days versus not.

Regarding age, 12% of participants were between 20 to 29, 28% were between 30 to 39, 28% were between 40 to 49, and 32% were between 50 and 60. The most prevalent racial and ethnic identity was non-Hispanic White followed by Hispanic and non-Hispanic Black. Regarding education, just under a third reported having a high school diploma or less, less than a quarter had some college, just under half had a college degree or more. Over two-thirds of respondents

**Table 1. Description of samples.**

| Variables | Full Sample (N = 3,593) | | | Drank in last 30 days (N = 2,429) | | |
|---|---|---|---|---|---|---|
| | Unweighted N | Weighted % or mean | (SE) | Unweighted N | Weighted % or mean | (SE) |
| Drinking to cope | | | | | | |
| Yes | 1,209 | 28.13% | | - | - | |
| No | 2,384 | 71.87% | | - | - | |
| Regularity (range 1–7) | - | - | | - | 3.05 | (0.10) |
| How Many (range 1–20) | - | - | | - | 2.30 | (0.11) |
| Sexual identity | | | | | | |
| Exclusively heterosexual | 1,987 | 96.54% | | 1,267 | 96.32% | |
| Exclusively gay/lesbian | 724 | 1.01% | | 535 | 1.11% | |
| Exclusively bisexual | 339 | 0.83% | | 240 | 0.90% | |
| Multi/another sexual identity | 543 | 1.62% | | 387 | 1.68% | |
| Gender identity | | | | | | |
| Cisgender man | 1,751 | 48.65% | | 1,240 | 51.59% | |
| Cisgender woman | 1,715 | 51.03% | | 1,107 | 48.15% | |
| Trans/another gender identity | 127 | 0.32% | | 82 | 0.26% | |
| Age Groups | | | | | | |
| 20–29 | 357 | 12.33% | | 263 | 13.79% | |
| 30–39 | 966 | 27.98% | | 685 | 28.15% | |
| 40–49 | 912 | 27.80% | | 614 | 28.39% | |
| 50–60 | 1,358 | 31.89% | | 867 | 29.67% | |
| Racial/ethnic Identity | | | | | | |
| Non-Hispanic White | 2,122 | 55.93% | | 1,456 | 54.53% | |
| Non-Hispanic Black | 309 | 7.44% | | 197 | 7.29% | |
| Non-Hispanic Asian | 200 | 6.71% | | 121 | 6.43% | |
| Hispanic | 568 | 21.67% | | 403 | 23.86% | |
| Non-Hispanic Multirace | 198 | 3.42% | | 137 | 3.68% | |
| Another | 196 | 4.83% | | 115 | 4.21% | |
| Education | | | | | | |
| High school or less | 626 | 31.00% | | 342 | 26.45% | |
| Some college | 698 | 22.27% | | 449 | 22.90% | |
| College degree | 2,269 | 46.73% | | 1,638 | 50.65% | |
| Employment Status | | | | | | |
| Not employed | 676 | 22.12% | | 430 | 20.29% | |
| Employed, not working | 108 | 2.63% | | 66 | 2.38% | |
| Employed part-time | 365 | 10.67% | | 225 | 9.95% | |
| Employed full-time | 2,444 | 64.58% | | 1,708 | 67.38% | |
| Marital status | | | | | | |
| Married | 2,645 | 80.72% | | 1,770 | 79.77% | |
| Cohabiting | 948 | 19.28% | | 659 | 20.23% | |
| Parenthood Status | | | | | | |
| No children | 2,665 | 69.36% | | 1844 | 70.60% | |
| Have children | 928 | 30.64% | | 585 | 29.40% | |
| Microaggressions (range 1–30) | - | 9.71 | (0.19) | - | 9.66 | (0.22) |
| Supportive climate (range 1–10) | - | 5.20 | (0.11) | - | 5.25 | (0.13) |
| COVID-19 Stress (range 4–20) | - | 10.53 | (0.23) | - | 10.78 | (0.28) |

(*Continued*)

**Table 1.** (Continued)

| | Full Sample (*N* = 3,593) | | | Drank in last 30 days (*N* = 2,429) | | |
|---|---|---|---|---|---|---|
| | Unweighted *N* | Weighted % or mean | (SE) | Unweighted *N* | Weighted % or mean | (SE) |
| COVID-19 Disruption (range 1–4) | - | 2.95 | (0.04) | - | 3.00 | (0.05) |

Source: National Couples' Health and Time Use Study

were employed full-time and just under a quarter were not employed. Eighty percent were married and 31% had children. Respondents reported an average of 10 microaggressions, a supportive climate score around the midpoint, a COVID-19 stress score of about 10, and COVID-19 disruption score of 3.0. These statistics were similar for the sample of drinkers.

Table 2 presents the results of stepwise logistic regression models that examined the relationship between sexual and gender identities and drinking alcohol to cope with the pandemic. Model 1 includes respondents' reports of their sexual and gender identities. Compared to heterosexual respondents, respondents who did not have a heterosexual sexual identity had significantly higher odds of drinking to cope (69% higher for exclusively gay/lesbian, 2.29 times higher for exclusively bisexual, and 62% higher for multiple or other sexual identities). Compared to cisgender men, the odds of drinking to cope were significantly lower (22%) for cisgender women. The difference in drinking to cope between people with transgender or another gender identity and cisgender men was not statistically significant. Following the addition of sociodemographic characteristics, the difference in drinking to cope between cisgender men and women was no longer statistically significant. There was significant variation in drinking to cope by race/ethnicity, education, and employment but not by age, relationship status, or parenthood status.

Model 3 included microaggressions and supportive climate, neither of which were associated with drinking to cope. Model 4 included COVID-19 stress and disruption. Whereas COVID-19 stress was not associated with drinking to cope, a one-point increase in COVID-19 disruption was associated with 30% higher odds of drinking to cope. Results concerning sexual and gender identities remained unchanged aside from a reemergence of a significant difference between cisgender women and men, and loss of significance between multi/another sexual identity and exclusively heterosexual. We also compared drinking alcohol to cope among all sexual and gender identity groups and there were no significant differences beyond those previously described (results not shown).

Table 3 includes models assessing associations between sexual and gender identities and regularity of drinking. People who identified as exclusively gay or lesbian engaged in more frequent drinking than did exclusively heterosexual people. Cisgender women drank significantly less frequently than did cisgender men. Microaggressions, supportive climate, or COVID-19 stress and disruption were not associated with regularity of drinking. Age, race/ ethnicity, education, and employment were associated with more regular drinking. We identified no other differences in drinking regularity among other sexual and gender identity groups (results not shown).

Table 4 includes models of the number of drinks per drinking occasion. In the baseline model, people who identified as exclusively bisexual consumed significantly more drinks than did exclusively heterosexual people, and cisgender men drank significantly more drinks than did cisgender women and those identifying as transgender or another gender identity. Number of drinks was associated with respondents' age, race/ethnicity, employment, and relationship status. Microaggressions and COVID-19 stress were associated with greater number of drinks consumed, but not supportive climate or COVID-19 disruption. The associations

**Table 2. Odds ratios predicting using alcohol to cope with the pandemic (N = 3,593).**

| Variables | Model 1 OR | Model 1 95% CI | Model 2 OR | Model 2 95% CI | Model 3 OR | Model 3 95% CI | Model 4 OR | Model 4 95% CI |
|---|---|---|---|---|---|---|---|---|
| Sexual identity (ref. exclusively heterosexual) | | | | | | | | |
| Exclusively gay/lesbian | 1.69 *** | (1.28, 2.22) | 1.53 ** | (1.14, 2.07) | 1.56 ** | (1.15, 2.11) | 1.50 ** | (1.10, 2.03) |
| Exclusively bisexual | 2.29 *** | (1.49, 3.50) | 2.26 *** | (1.42, 3.60) | 2.29 *** | (1.43, 3.66) | 2.14 *** | (1.34, 3.43) |
| Multi/another sexual identity | 1.62 * | (1.08, 2.43) | 1.57 * | (1.00, 2.46) | 1.57 * | (1.01, 2.46) | 1.49 | (0.95, 2.34) |
| Gender identity (ref. cisgender man) | | | | | | | | |
| Cisgender woman | 0.78 * | (0.63, 0.97) | 0.82 | (0.65, 1.04) | 0.83 | (0.65, 1.05) | 0.78 * | (0.62, 1.00) |
| Trans/another gender identity | 0.75 | (0.35, 1.64) | 0.79 | (0.35, 1.75) | 0.77 | (0.35, 1.73) | 0.75 | (0.34, 1.66) |
| Age (ref. 20–29) | | | | | | | | |
| 30–39 | - | | 1.32 | (0.85, 2.05) | 1.33 | (0.86, 2.06) | 1.31 | (0.84, 2.04) |
| 40–49 | - | | 1.31 | (0.84, 2.04) | 1.32 | (0.85, 2.06) | 1.28 | (0.82, 2.01) |
| 50–60 | - | | 1.04 | (0.65, 1.66) | 1.07 | (0.67, 1.70) | 1.03 | (0.64, 1.64) |
| Race/ethnic identity (ref. Non-Hispanic White) | | | | | | | | |
| Non-Hispanic Black | - | | 0.77 | (0.54, 1.09) | 0.72 | (0.50, 1.04) | 0.72 | (0.50, 1.04) |
| Non-Hispanic Asian | - | | 0.56 * | (0.33, 0.95) | 0.55 * | (0.33, 0.93) | 0.53 * | (0.32, 0.90) |
| Hispanic | - | | 0.90 | (0.67, 1.21) | 0.90 | (0.67, 1.21) | 0.88 | (0.65, 1.18) |
| Non-Hispanic Multirace | - | | 0.89 | (0.55, 1.45) | 0.88 | (0.54, 1.43) | 0.83 | (0.51, 1.35) |
| Another | - | | 0.86 | (0.53, 1.39) | 0.85 | (0.53, 1.38) | 0.85 | (0.53, 1.37) |
| Education (ref. high school or less) | | | | | | | | |
| Some college | - | | 1.44 * | (1.03, 2.00) | 1.45 * | (1.04, 2.03) | 1.36 | (0.97, 1.91) |
| College degree | - | | 1.80 *** | (1.32, 2.45) | 1.83 *** | (1.34, 2.50) | 1.59 ** | (1.16, 2.19) |
| Employment status (ref. not employed) | | | | | | | | |
| Employed, not working | - | | 0.82 | (0.40, 1.68) | 0.81 | (0.39, 1.67) | 0.77 | (0.37, 1.61) |
| Employed part-time | - | | 1.21 | (0.78, 1.87) | 1.22 | (0.79, 1.89) | 1.24 | (0.79, 1.93) |
| Employed full-time | - | | 1.41 * | (1.06, 1.89) | 1.42 * | (1.06, 1.90) | 1.52 ** | (1.13, 2.06) |
| Married (ref. cohabiting) | - | | 0.74 * | (0.56, 0.98) | 0.76 | (0.57, 1.01) | 0.78 | (0.59, 1.05) |
| Parenthood status (ref. no children) | - | | 1.15 | (0.88, 1.51) | 1.15 | (0.88, 1.51) | 1.20 | (0.92, 1.57) |
| Microaggressions | - | | - | | 1.02 | (0.99, 1.05) | 1.01 | (0.98, 1.04) |
| Supportive climate | - | | - | | 1.00 | (0.95, 1.06) | 1.01 | (0.96, 1.06) |
| COVID-19 Stress | - | | - | | - | | 1.02 | (0.99, 1.04) |
| COVID-19 Disruption | - | | - | | - | | 1.30 *** | (1.10, 1.52) |

Source: National Couples' Health and Time Use Study

Note: OR = odds ratio; CI = confidence interval; Ref. = reference. Month of survey is included in all models, coefficients not reported.

\* $p < .05$

\*\* $p < .01$

\*\*\* $p < .001$

between sexual and gender identities remain unchanged, and no other differences among LGBTQ+ groups were observed (results not shown).

In other analyses not shown, we found that COVID-19 stress, COVID-19 disruption, microaggressions, and climate were related to sexual identity and gender identity. Across samples, people with an exclusively gay/lesbian, exclusively bisexual, and multi/another sexual identity scored significantly higher on COVID-19 stress and disruption, as did cisgender woman and trans/another gender identity in relation to cisgender man. People with a multi/another sexual identity scored significantly lower on climate than did exclusively heterosexual people, as did those with a trans-another gender identity in relation to cisgender man. Those

**Table 3. Coefficients from regression models predicting regularity of drinking ($N = 2,429$).**

| Variables | Model 1 Coefficient | | 95% CI | | Model 2 Coefficient | | 95% CI | | Model 3 Coefficient | | 95% CI | | Model 4 Coefficient | | 95% CI | |
|---|---|---|---|---|---|---|---|---|---|---|---|---|---|---|---|---|
| Sexual identity (ref. exclusively heterosexual) | | | | | | | | | | | | | | | | |
| Exclusively gay/lesbian | 0.523 | *** | (0.25, | 0.80) | 0.474 | ** | (0.18, | 0.77) | 0.467 | ** | (0.17, | 0.76) | 0.469 | ** | (0.17, | 0.76) |
| Exclusively bisexual | 0.213 | | (-0.22, | 0.64) | 0.268 | | (-0.14, | 0.68) | 0.262 | | (-0.15, | 0.68) | 0.259 | | (-0.16, | 0.68) |
| Multi/another sexual identity | 0.167 | | (-0.22, | 0.55) | 0.221 | | (-0.18, | 0.62) | 0.220 | | (-0.18, | 0.62) | 0.231 | | (-0.18, | 0.64) |
| Gender identity (ref. cisgender man) | | | | | | | | | | | | | | | | |
| Cisgender woman | -0.438 | *** | (-0.64, | -0.23) | -0.399 | *** | (-0.61, | -0.18) | -0.402 | *** | (-0.62, | -0.19) | -0.392 | *** | (-0.61, | -0.17) |
| Trans/another gender identity | -0.268 | | (-1.01, | 0.48) | -0.276 | | (-0.98, | 0.42) | -0.280 | | (-0.98, | 0.42) | -0.275 | | (-0.98, | 0.43) |
| Age (ref. 20–29) | | | | | | | | | | | | | | | | |
| 30–39 | - | | | | 0.381 | * | (0.03, | 0.73) | 0.377 | * | (0.02, | 0.73) | 0.372 | * | (0.02, | 0.73) |
| 40–49 | - | | | | 0.416 | * | (0.06, | 0.77) | 0.412 | * | (0.06, | 0.76) | 0.405 | * | (0.05, | 0.76) |
| 50–60 | - | | | | 0.257 | | (-0.14, | 0.65) | 0.250 | | (-0.14, | 0.64) | 0.242 | | (-0.15, | 0.63) |
| Race/ethnic identity (ref. Non-Hispanic White) | | | | | | | | | | | | | | | | |
| Non-Hispanic Black | - | | | | -0.349 | * | (-0.69, | -0.00) | -0.331 | | (-0.68, | 0.02) | -0.326 | | (-0.68, | 0.03) |
| Non-Hispanic Asian | - | | | | -0.455 | | (-0.92, | 0.01) | -0.445 | | (-0.91, | 0.02) | -0.446 | | (-0.91, | 0.02) |
| Hispanic | - | | | | -0.378 | ** | (-0.63, | -0.13) | -0.378 | ** | (-0.63, | -0.13) | -0.374 | ** | (-0.63, | -0.12) |
| Non-Hispanic Multirace | - | | | | -0.212 | | (-0.65, | 0.22) | -0.210 | | (-0.64, | 0.22) | -0.209 | | (-0.61, | 0.23) |
| Another | - | | | | -0.325 | | (-0.79, | 0.14) | -0.321 | | (-0.79, | 0.15) | -0.330 | | (-0.79, | 0.14) |
| Education (ref. high school or less) | | | | | | | | | | | | | | | | |
| Some college | - | | | | 0.047 | | (-0.26, | 0.35) | 0.043 | | (-0.26, | 0.35) | 0.044 | | (-0.26, | 0.35) |
| College degree | - | | | | 0.316 | * | (0.02, | 0.61) | 0.309 | * | (0.02, | 0.60) | 0.306 | * | (0.01, | 0.60) |
| Employment status (ref. not employed) | - | | | | | | | | | | | | | | | |
| Employed, not working | | | | | -0.106 | | (-0.68, | 0.46) | -0.105 | | (-0.67, | 0.46) | -0.091 | | (-0.66, | 0.48) |
| Employed part-time | | | | | 0.371 | | (-0.05, | 0.79) | 0.374 | | (-0.05, | 0.80) | 0.387 | | (-0.04, | 0.81) |
| Employed full-time | - | | | | 0.283 | * | (0.03, | 0.54) | 0.286 | * | (0.03, | 0.54) | 0.299 | * | (0.04, | 0.56) |
| Married (ref. cohabiting) | - | | | | -0.100 | | (-0.37, | 0.17) | -0.108 | | (-0.38, | 0.17) | -0.118 | | (-0.39, | 0.16) |
| Parenthood status (ref. no children) | - | | | | 0.286 | * | (0.02, | 0.55) | 0.285 | * | (0.02, | 0.55) | 0.284 | * | (0.02, | 0.55) |
| Microaggressions | - | | | | - | | | | -0.007 | | (-0.03, | 0.02) | -0.006 | | (-0.03, | 0.02) |
| Supportive climate | - | | | | - | | | | -0.007 | | (-0.06, | 0.04) | -0.010 | | (-0.06, | 0.04) |
| COVID-19 Stress | - | | | | - | | | | - | | | | -0.011 | | (-0.04, | 0.01) |
| COVID-19 Disruption | - | | | | - | | | | - | | | | 0.050 | | (-0.09, | 0.19) |

Source: National Couples' Health and Time Use Study

Note: CI = confidence interval; Ref. = reference. Month of survey is included in all models, coefficients not reported.

\* $p < .05$

\*\* $p < .01$

\*\*\* $p < .001$

with an exclusively gay/lesbian identity and those with a multi/another sexual identity scored significantly higher on microaggressions than did exclusively heterosexual people, as did those with a trans/another gender identity in relation to cisgender man.

## Sensitivity analysis

We conducted supplemental analysis to gauge the robustness of our results. We categorized non-cisgender, non-heterosexual identities in different ways. For example, we combined respondents who were exclusively heterosexual, gay/lesbian, or bisexual, with those who were heterosexual, gay/lesbian, or bisexual, plus one or more other identities, and there were no

**Table 4. Coefficients from regression models predicting number of drinks (N = 2,429).**

| Variables | Model 1 Coefficient | Model 1 95% CI | Model 2 Coefficient | Model 2 95% CI | Model 3 Coefficient | Model 3 95% CI | Model 4 Coefficient | Model 4 95% CI |
|---|---|---|---|---|---|---|---|---|
| Sexual identity (ref. exclusively heterosexual) | | | | | | | | |
| Exclusively gay/lesbian | 0.042 | (-0.09, 0.18) | 0.026 | (-0.11, 0.16) | 0.047 | (-0.09, 0.18) | 0.036 | (-0.10, 0.17) |
| Exclusively bisexual | 0.288 * | (0.04, 0.53) | 0.254 * | (0.00, 0.50) | 0.277 * | (0.02, 0.53) | 0.261 * | (0.01, 0.51) |
| Multi/another sexual identity | 0.009 | (-0.15, 0.16) | 0.005 | (-0.16, 0.17) | -0.002 | (-0.16, 0.15) | -0.036 | (-0.19, 0.11) |
| Gender identity (ref. cisgender man) | | | | | | | | |
| Cisgender woman | -0.320 *** | (-0.42, -0.22) | -0.311 *** | (-0.41, -0.22) | -0.304 *** | (-0.40, -0.21) | -0.332 *** | (-0.43, -0.23) |
| Trans/another gender identity | -0.270 * | (-0.50, -0.04) | -0.278 * | (-0.53, -0.03) | -0.292 * | (-0.53, -0.06) | -0.306 * | (-0.54, -0.07) |
| Age (ref. 20–29) | | | | | | | | |
| 30–39 | - | | 0.140 | (-0.01, 0.29) | 0.153 * | (0.00, 0.30) | 0.155 * | (0.00, 0.31) |
| 40–49 | - | | 0.130 | (-0.04, 0.30) | 0.147 | (-0.02, 0.31) | 0.151 | (-0.02, 0.32) |
| 50–60 | - | | 0.116 | (-0.06, 0.29) | 0.148 | (-0.03, 0.32) | 0.142 | (-0.03, 0.31) |
| Race/ethnic identity (ref. Non-Hispanic White) | | | | | | | | |
| Non-Hispanic Black | - | | 0.006 | (-0.20, 0.22) | -0.057 | (-0.27, 0.15) | -0.072 | (-0.28, 0.13) |
| Non-Hispanic Asian | - | | -0.497 *** | (-0.71, -0.29) | -0.515 *** | (-0.73, -0.30) | -0.522 *** | (-0.73, -0.31) |
| Hispanic | - | | -0.080 | (-0.21, 0.05) | -0.082 | (-0.21, 0.04) | -0.105 | (-0.23, 0.02) |
| Non-Hispanic Multirace | - | | -0.272 ** | (-0.46, -0.09) | -0.289 ** | (-0.47, -0.10) | -0.309 *** | (-0.50, -0.12) |
| Another | - | | -0.278 ** | (-0.48, -0.08) | -0.295 ** | (-0.49, -0.10) | -0.280 ** | (-0.47, -0.09) |
| Education (ref. high school or less) | | | | | | | | |
| Some college | - | | 0.115 | (-0.02, 0.25) | 0.124 | (-0.01, 0.26) | 0.101 | (-0.03, 0.23) |
| College degree | - | | -0.054 | (-0.18, 0.07) | -0.030 | (-0.16, 0.09) | -0.067 | (-0.19, 0.06) |
| Employment status (ref. not employed) | | | | | | | | |
| Employed, not working | - | | 0.121 | (-0.41, 0.65) | 0.105 | (-0.44, 0.65) | 0.075 | (-0.48. 0.63) |
| Employed part-time | - | | -0.176 * | (-0.32, -0.04) | -0.179 ** | (-0.32, -0.04) | -0.194 ** | (-0.33, -0.06) |
| Employed full-time | - | | -0.034 | (-0.15, 0.08) | -0.030 | (-0.14, 0.08) | -0.032 | (-0.15, 0.08) |
| Married (ref. cohabiting) | - | | -0.114 * | (-0.22, -0.01) | -0.090 | (-0.20, 0.02) | -0.068 | (-0.18, 0.04) |
| Parenthood status (ref. no children) | - | | 0.004 | (-0.11, 0.12) | 0.004 | (-0.11, 0.12) | 0.017 | (-0.10, 0.13) |
| Microaggressions | - | | - | | 0.019 ** | (0.01, 0.03) | 0.016 * | (0.00, 0.03) |
| Supportive climate | - | | - | | -0.003 | (-0.03, 0.02) | 0.003 | (-0.02, 0.03) |
| COVID-19 Stress | - | | - | | - | | 0.020 *** | (0.01, 0.03) |
| COVID-19 Disruption | - | | - | | - | | -0.004 | (-0.06, 0.05) |

Source: National Couples' Health and Time Use Study

Note: CI = confidence interval; Ref. = reference. Month of survey is included in all models, coefficients not reported.

* $p < .05$

** $p < .01$

*** $p < .001$

significant differences in effects. Moreover, models were repeated comparing the full sample (including those who did not drink in the last 30 days) and the smaller sample (drank in the last 30 days). Results regarding drinking to cope based on the sample restricted to drinkers were similar to those based on the broader sample, except that there was no longer a difference between cisgender men and cisgender women, suggesting that men and women who are drinkers were similarly likely to use alcohol to cope. Results concerning regularity that include non-drinkers were similar except that people who identified as transgender or another gender identity drank significantly less often than did cisgender men. Findings regarding number of drinks consumed were similar between samples except that the positive effect of microaggressions was no longer statistically significant when the sample included non-drinkers.

We also ran logit models using two measures of binge drinking: (a) 4+ drinks (standard for women), and (b) 5+ drinks (standard for men), among people who drank in last 30 days. Our results with respect to gender identity mirrored our findings with respect to number of drinks. Cisgender women had lower odds of binge drinking than cisgender men (both measures). Transgender/another identity had lower odds of binge drinking (5+ drinks) relative to cisgender men. Sexual identity was not associated with binge drinking. We tested the validity of our alcohol measures. Drinking to cope was positively correlated with regularity of drinking and number of drinks consumed. Having a family member, friend, or doctor express worry about the respondent's alcohol consumption was associated with increased odds of drinking to cope, more regular drinking, and drinking in greater amounts (results not shown).

## Discussion

Social, emotional, and physical health have declined since the onset of the pandemic, and the use of alcohol and other substances has increased. Previous national emergencies in the United States (e.g., 9/11 and Hurricane Katrina), had enduring negative effects on mental and physical health, including increased alcohol consumption [77–79]. Few studies have examined alcohol use among the LGBTQ+ population since the pandemic and in the face of such disasters. Our findings are based on a U.S. nationally representative sample conducted at the height of the pandemic, with an oversample LGBTQ+ people, rich measures of sexual and gender identities, multiple measures of alcohol use, COVID-19-specific stress and disruption measures, measures tapping minority stress, and an array of sociodemographic variables. Exclusively gay, lesbian, or bisexual persons had significantly higher odds of drinking to cope than did exclusively heterosexual persons, as did cisgender men compared to cisgender women. Gay or lesbian persons drank significantly more frequently than did heterosexual persons, and cisgender women drank less frequently than did cisgender men. Exclusively bisexual people drank significantly more drinks than exclusively heterosexual people, cisgender men consumed more drinks than cisgender women and people who identified as transgender or another gender identity. Our findings were robust across models and were not explained by socioeconomic factors, minority stress indicators, or COVID-19 stress and disruption.

Our results are consistent with minority stress theory and previous research on the particularly negative impact of the pandemic among LGBTQ+ people, as indicated by greater alcohol use among some groups, higher scores on microaggressions, COVID-19 stress and disruption, and lower scores on climate [20–22, 26]. However, because we identified differences in drinking patterns *between* LGBTQ+ groups, researchers should avoid making blanket statements about alcohol use and this population. Our findings did not support the hypothesis that minority stress and COVID-19-stress and disruption, while associated with some measures of alcohol use, explain differences in alcohol use by sexual and gender identity. LGBTQ+ persons may drink more for reasons not measured such as family histories, individual factors, and greater social isolation [80, 81].

### Limitations

Although we provided new insights into alcohol use during the pandemic, our study has limitations. NCHAT is cross-sectional and data were collected during the early period of the pandemic, providing only a snapshot of alcohol use. Our findings may reflect longstanding differences in alcohol consumption between people of different sexual and gender identities [82]. We also do not know if the differences we observed were sustained beyond the pandemic. Our measures of alcohol use were limited to self-reports and were subject to potential recall error, and only drinking in the last 30 days was included. The data were restricted to people

between the ages of 20 and 60. We also do not account for the respondent's partner's drinking behavior [83, 84], or include singles who may have been more stressed [85]. Our sample of non-cisgender people was also small. Understanding alcohol use among LGBTQ+ populations is also hampered by lack of tests of validity of standard measures of alcohol consumption and lack of consensus as to handle gender-based measures such as with binge drinking [86].

## Conclusion

The COVID-19 pandemic has altered the way we live in fundamental ways: the way we work, interact with others, perform our daily routines, and engage in behaviors related to our health and well-being. Our work focused on alcohol use and important next steps will incorporate indicators of drug use and misuse. Moving forward, researchers will need to continuously assess these associations, as sources of discrimination and stress will persist beyond the pandemic—the alarming rise in anti-LGBTQ+ legislation in the U.S. is evidence of this [87]. Although LGBTQ+ people have disproportionate sources of stress, they varied in their response to the pandemic, at least in terms of drinking behavior. The well-being of individuals after collective trauma has the potential to rebound relatively quickly [88] with long-term community support, consistent medical care, and mental health support services [89, 90]. However, only 7% of substance use treatment programs offer specialized services for LGBTQ + people. They also enter treatment with more severe AUDs and are more likely to have co-morbid psychiatric disorders [91]. This lack of care was exacerbated by the pandemic because people were often cut-off from counselors and support services [39]. LGBTQ+ populations benefit from programs that address transphobia/homophobia, difficult family relationships, and social isolation [92]. Potential sources of resilience among sexual and gender diverse individuals should be explored.

## Author Contributions

**Conceptualization:** Susan D. Stewart, Wendy D. Manning, Kristen E. Gustafson, Claire Kamp Dush.

**Formal analysis:** Kristen E. Gustafson.

**Funding acquisition:** Wendy D. Manning.

**Investigation:** Wendy D. Manning.

**Methodology:** Susan D. Stewart, Wendy D. Manning, Kristen E. Gustafson, Claire Kamp Dush.

**Project administration:** Susan D. Stewart.

**Writing – original draft:** Susan D. Stewart, Wendy D. Manning, Kristen E. Gustafson, Claire Kamp Dush.

**Writing – review & editing:** Susan D. Stewart, Wendy D. Manning, Kristen E. Gustafson, Claire Kamp Dush.

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
