## [Decision Letter · Decision Letter 0]

1 Jul 2024

PONE-D-24-05598Sexual and gender identities and alcohol use during the COVID-19 pandemicPLOS ONE

Dear Dr. Stewart,

Thank you for submitting your manuscript to PLOS ONE. After careful consideration, we feel that it has merit but does not fully meet PLOS ONE’s publication criteria as it currently stands. Therefore, we invite you to submit a revised version of the manuscript that addresses the points raised during the review process.

We look forward to receiving your revised manuscript.

Kind regards,

Hong-Van Tieu

Academic Editor

PLOS ONE

Additional Editor Comments:

Discussion section should be expanded on.

For example, the findings that "Gay or lesbian persons drank significantly more frequently than did heterosexual persons, and cisgender women drank less frequently than did cisgender men. These findings are consistent with previous research." More context can be provided with prior research findings, with references cited.

Please check grammar in the paper. For example, in the discussion, "measure" should be plural: multiple measure of alcohol use

Describe why other substance use, besides alcohol use, was not explored in the analysis

Reviewers' comments:

Reviewer's Responses to Questions

**Comments to the Author**

1. Is the manuscript technically sound, and do the data support the conclusions?

Reviewer #1: Yes

2. Has the statistical analysis been performed appropriately and rigorously? 

Reviewer #1: Yes

3. Have the authors made all data underlying the findings in their manuscript fully available?

Reviewer #1: Yes

4. Is the manuscript presented in an intelligible fashion and written in standard English?

Reviewer #1: Yes

5. Review Comments to the Author

Reviewer #1: A.

I think it is a very interesting paper that is assessing the relationship between alcohol use and gender identity. And having the added dimensions of minority stressor and support factors as well as the Covid-specific measures

provides important insight into the potential dynamics with alcohol use. There are some statistical questions that may help enhance the results as well as some sections of text which need clarification/reconciliation with the tables.

B.

Major issues:

1. The early time period of the study data may be qualitatively different than the later time period of the study data so it might be useful to examine potential time period effect. In the text and as a footnote there is a reference to the Month of the interview as a covariate, but I was unsure of the finding.

2. I think it would be interesting to include the results for Minority Stress, Supportive climate, Covid Stress and Covid Disruption by Sex ID and Gender ID.

3. You used OLS for the regularity and number of drinks outcomes but did these measures follow a normal distribution?

4. You have number of drinks, but it might be informative to add a Binge Drinking outcome even if there isn't a standard definition.

5. You found Age to not be associated but it might be useful to examine bivariate associations between Age cohorts and your outcomes for potential non-linear relationships.

Minor issues:

1. Line 253 - based on your description of the measure and the mean value=3.1, doesn't the mean value correspond to the 1 or 2 days a week response category

2. Line 259 seems like you need to add "Results not shown" for clarity.

2. Line 260 need accompanying SE

3. Line 264 The "80% have children" needs to be fixed so that it matches the table.

4. Table 1 For the Age row you have "mean" in the label column, but it should be "range"

5. Line 275 is not clear to me because the Odds for Transgender/another gender identity is not similar to cisgender man, it is similar to cisgender woman, but the effect was not significant

6. Line 289 I am not sure what this means b/c Model 1 indicates that there is a significant difference in drinking alcohol to cope between sexual identity groups

7. Table 4 Employed part-time seems like the coefficient should be negative

8. Line 319 You are saying full sample but then saying those that did not drink in the last 30 days isn't that a subset of the full sample, please clarify this sentence.

C. Misc. remarks (Other points)

None

6. PLOS authors have the option to publish the peer review history of their article (what does this mean?). If published, this will include your full peer review and any attached files.

Reviewer #1: No

---

## [Author Response · Author response to Decision Letter 0]

19 Jul 2024

Journal Requirements:

We have reviewed these templates and our manuscript is in this format.

This has been corrected. 

We added the following statement in the Methods section: “The study was approved by The Ohio State Behavioral and Social Sciences IRB (#2018B0246) and The University of Minnesota (STUDY00013894). Consent was established at the beginning of the survey. Respondents were shown and reviewed the fully informed consent, and the survey only continued if positive consent was obtained by selecting ‘I consent.’”

We have reviewed our references and they are complete and correct with no changes. 

Additional Editor Comments:

(1) Discussion section should be expanded on.

For example, the findings that "Gay or lesbian persons drank significantly more frequently than did heterosexual persons, and cisgender women drank less frequently than did cisgender men. These findings are consistent with previous research." More context can be provided with prior research findings, with references cited.

We have omitted this sentence, because unlike most previous research, we break LGBTQ+ into very specific identities and therefore provide a more nuanced understanding than prior studies. Therefore, we state:

“Our results are consistent with minority stress theory and previous research on the particularly negative impact of the pandemic among LGBTQ+ people, as indicated by greater alcohol use among some groups, higher scores on microaggressions, COVID-19 stress and disruption, and lower scores on climate (Hughes et al. 2020; Kneale and Bécares 2021; Manning and Kamp Dush, 2022; Schuler et al. 2020). However, because we identified differences in drinking patterns between LGBTQ+ groups, researchers should avoid making blanket statements about alcohol use and this population.”

(2) Please check grammar in the paper. For example, in the discussion, "measure" should be plural: multiple measure of alcohol use

Thank you. We have checked again for errors and have made corrections. 

(3) Describe why other substance use, besides alcohol use, was not explored in the analysis

We have carefully considered expanding our analysis to other substances and have decided to continue to limit our analysis to alcohol use for several reasons. Alcohol is normative and legal and most other drugs are illegal and is associated with greater societal disapproval. Previous studies of LGBTQ+ populations have shown different sets of variables are associated with the use of alcohol versus other drugs. Our measures are also not consistent across substances. We had noted in our discussion that examining drug use is an important next step.

Reviewer's Responses to Questions

Reviewer #1:

A. I think it is a very interesting paper that is assessing the relationship between alcohol use and gender identity. And having the added dimensions of minority stressor and support factors as well as the Covid-specific measures provides important insight into the potential dynamics with alcohol use. There are some statistical questions that may help enhance the results as well as some sections of text which need clarification/reconciliation with the tables.

We very much appreciate this reviewer’s careful reading of our paper and have responded in detail to each of their concerns below. 

B. Major issues:

1. The early time period of the study data may be qualitatively different than the later time period of the study data so it might be useful to examine potential time period effect. In the text and as a footnote there is a reference to the Month of the interview as a covariate, but I was unsure of the finding.

That is correct. We did include month of interview in our analysis but did not include it in the tables. The month of interview did not have a significant effect on any of our measures of alcohol use. This is now noted in the text: “Month of interview was not associated with drinking to cope, regularity, or amount consumed (results not shown).”

2. I think it would be interesting to include the results for Minority Stress, Supportive climate, Covid Stress and Covid Disruption by Sex ID and Gender ID.

We agree. In the initial submission, we had included two statements about these relationships but did not show our results. We realize these statements could have been clearer. Therefore, we clarified how COVID-19 stress, COVID-19 disruption, microaggressions, climate, were related to sexual identity and gender identity. We have included the following statements in the revision:

Results Section: “In other analyses not shown, we found that COVID-19 stress, COVID-19 disruption, microaggressions, and climate were related to sexual identity and gender identity. Across samples, people with an exclusively gay/lesbian, exclusively bisexual, and multi/another sexual identity scored significantly higher on COVID-19 stress and disruption, as did cisgender woman and trans/another gender identity in relation to cisgender man. People with a multi/another sexual identity scored significantly lower on climate than did exclusively heterosexual people, as did those with a trans-another gender identity in relation to cisgender man. Those with an exclusively gay/lesbian identity and those with a multi/another sexual identity scored significantly higher on microaggressions than did exclusively heterosexual people, as did those with a trans/another gender identity in relation to cisgender man.”

Discussion Section: “Our results are consistent with minority stress theory and previous research on the particularly negative impact of the pandemic on LGBTQ+ people as indicated by their higher scores on microaggressions, COVID-19 stress and disruption, and lower scores on climate.” 

3. You used OLS for the regularity and number of drinks outcomes but did these measures follow a normal distribution?

We thank this reviewer for their attention to this. We checked the distributions of both of these variables. Both are skewed but only number of drinks meets the threshold for a negative binomial model (i.e., the variance is higher than the mean). We reran the analysis using this model. Only one coefficient, for exclusively bisexual, was meaningfully changed in size or significance. This is noted in the text. 

4. You have number of drinks, but it might be informative to add a Binge Drinking outcome even if there isn't a standard definition.

We agree. We ran logit models using two measures of binge drinking: (a) 4+ drinks (standard for women), and (b) 5+ drinks (standard for men), among people who drank in last 30 days. Our results with respect to gender identity mirrored our findings with respect to number of drinks. Cisgender women had lower odds of binge drinking than cisgender men (both measures). Transgender/another identity had lower odds of binge drinking (5+ drinks) relative to cisgender men. Sexual identity was not associated with binge drinking. We have noted our binge drinking findings in the text. We added this text to the Results. 

5. You found Age to not be associated but it might be useful to examine bivariate associations between Age cohorts and your outcomes for potential non-linear relationships.

We checked the effect of age cohort by replacing mean age with categories of age. No age category was associated with drinking to cope (ref. group 20-29), similar to our previous findings. We had reported that mean age was not associated with regularity or number of drinks—this has been revised. People age 30-39 and 40-49 drank significantly less regularly than people age 20-29. With respect to number of drinks, people age 30-39 drank significantly fewer drinks than people age 20-29. 

C. Minor issues:

1. Line 253 - based on your description of the measure and the mean value=3.1, doesn't the mean value correspond to the 1 or 2 days a week response category

Thank you. This has been corrected in the text. 

2. Line 259 seems like you need to add "Results not shown" for clarity.

Thank you. This has been added. 

3. Line 260 need accompanying SE 

We are now using categories of age. 

4. Line 264 The "80% have children" needs to be fixed so that it matches the table.

The text has been corrected to read “31%.” 

5. Table 1 For the Age row you have "mean" in the label column, but it should be "range"

We are now using categories of age. 

6. Line 275 is not clear to me because the Odds for Transgender/another gender identity is not similar to cisgender man, it is similar to cisgender woman, but the effect was not significant

How we worded this was confusing. We have revised the text to say, “Compared to cisgender men, the odds of drinking to cope were significantly lower (22%) for cisgender women. The difference in drinking to cope between people with transgender or another gender identity and cisgender men was not statistically significant.”

7. Line 289 I am not sure what this means b/c Model 1 indicates that there is a significant difference in drinking alcohol to cope between sexual identity groups

We agree this text was confusing. We have revised the text to say, “We also compared drinking alcohol to cope among all sexual and gender identity groups and there were no significant differences beyond those previously described (results not shown).”

8. Table 4 Employed part-time seems like the coefficient should be negative

That is correct. This has been fixed. 

9. Line 319 You are saying full sample but then saying those that did not drink in the last 30 days isn't that a subset of the full sample, please clarify this sentence.

We apologize for the error. The sentence has been revised to read, “Moreover, models were repeated comparing the full sample (including those who did not drink in the last 30 days) and the smaller sample (drank in the last 30 days).”

Journal Requirements:

We have reviewed these templates and our manuscript is in this format.

This has been corrected. 

We added the following statement in the Methods section: “The study was approved by The Ohio State Behavioral and Social Sciences IRB (#2018B0246) and The University of Minnesota (STUDY00013894). Consent was established at the beginning of the survey. Respondents were shown and reviewed the fully informed consent, and the survey only continued if positive consent was obtained by selecting ‘I consent.’”

We have reviewed our references and they are complete and correct with no changes. 

Additional Editor Comments:

(1) Discussion section should be expanded on.

For example, the findings that "Gay or lesbian persons drank significantly more frequently than did heterosexual persons, and cisgender women drank less frequently than did cisgender men. These findings are consistent with previous research." More context can be provided with prior research findings, with references cited.

We have omitted this sentence, because unlike most previous research, we break LGBTQ+ into very specific identities and therefore provide a more nuanced understanding than prior studies. Therefore, we state:

“Our results are consistent with minority stress theory and previous research on the particularly negative impact of the pandemic among LGBTQ+ people, as indicated by greater alcohol use among some groups, higher scores on microaggressions, COVID-19 stress and disruption, and lower scores on climate (Hughes et al. 2020; Kneale and Bécares 2021; Manning and Kamp Dush, 2022; Schuler et al. 2020). However, because we identified differences in drinking patterns between LGBTQ+ groups, researchers should avoid making blanket statements about alcohol use and this population.”

(2) Please check grammar in the paper. For example, in the discussion, "measure" should be plural: multiple measure of alcohol use

Thank you. We have checked again for errors and have made corrections. 

(3) Describe why other substance use, besides alcohol use, was not explored in the analysis

We have carefully considered expanding our analysis to other substances and have decided to continue to limit our analysis to alcohol use for several reasons. Alcohol is normative and legal and most other drugs are illegal and is associated with greater societal disapproval. Previous studies of LGBTQ+ populations have shown different sets of variables are associated with the use of alcohol versus other drugs. Our measures are also not consistent across substances. We had noted in our discussion that examining drug use is an important next step.

Reviewer's Responses to Questions

Reviewer #1:

A. I think it is a very interesting paper that is assessing the relationship between alcohol use and gender identity. And having the added dimensions of minority stressor and support factors as well as the Covid-specific measures provides important insight into the potential dynamics with alcohol use. There are some statistical questions that may help enhance the results as well as some sections of text which need clarification/reconciliation with the tables.

We very much appreciate this reviewer’s careful reading of our paper and have responded in detail to each of their concerns bel

---

## [Editor Report · Decision Letter 1]

29 Jul 2024

PONE-D-24-05598R1Sexual and gender identities and alcohol use during the COVID-19 pandemicPLOS ONE

Dear Dr. Stewart,

Thank you for submitting your manuscript to PLOS ONE. After careful consideration, we feel that it has merit but does not fully meet PLOS ONE’s publication criteria as it currently stands. Therefore, we invite you to submit a revised version of the manuscript that addresses the points raised during the review process.

We look forward to receiving your revised manuscript.

Kind regards,

Hong-Van Tieu

Academic Editor

PLOS ONE

Journal Requirements:

Additional Editor Comments:

Thank you for responding to the comments and making the requested edits. Please see below for minor revisions:

Please review and make spelling checks throughout manuscript.

Some noted corrections needed below:

Line 187, spelling correction for COVID-19

Line 198 add “years” to Age ranged from 20 to 60

Line 200 were surveys completed in Spanish included in the analysis?

..were collected on-line in Spanish and English from September 2020 to April 2021..

Line 242 check paragraph indentation

Line 387 there is an extra period. cisgender men. .

---

## [Author Response · Author response to Decision Letter 1]

31 Jul 2024

Please advise as to how to fix this issue.

---

## [Editor Report · Decision Letter 2]

2 Aug 2024

Sexual and gender identities and alcohol use during the COVID-19 pandemic

PONE-D-24-05598R2

Dear Dr. Stewart,

We’re pleased to inform you that your manuscript has been judged scientifically suitable for publication and will be formally accepted for publication once it meets all outstanding technical requirements.

Kind regards,

Hong-Van Tieu

Academic Editor

PLOS ONE
---

## [Editor Report · Acceptance letter]

29 Aug 2024

PONE-D-24-05598R2 

PLOS ONE

Dear Dr. Stewart, 

I'm pleased to inform you that your manuscript has been deemed suitable for publication in PLOS ONE. Congratulations! Your manuscript is now being handed over to our production team.

Kind regards, 

on behalf of

Dr. Hong-Van Tieu 

Academic Editor

PLOS ONE